# The Pharmacogenetics of Cannabis in the Treatment of Chronic Pain

**DOI:** 10.3390/genes13101832

**Published:** 2022-10-11

**Authors:** Paolo Poli, Luana Peruzzi, Pierdomenico Maurizi, Antonella Mencucci, Antonella Scocca, Simona Carnevale, Ottavia Spiga, Annalisa Santucci

**Affiliations:** 1POLIPAIN CLINIC, SIRCA Italian Society of Cannabis Research, 56124 Pisa, Italy; 2Department of Chemical Biotechnology and Pharmacy, Siena University, 53100 Siena, Italy; 3Occupational Medicine and Palliative Care Unit, Arezzo Hospital, 52100 Arezzo Tuscany, Italy

**Keywords:** cannabis therapy, genetic polymorphisms, pharmacokinetics of cannabis, pain treatment

## Abstract

Background: The increase in the medical use of cannabis has revealed a number of beneficial effects, a variety of adverse side effects and great inter-individual variability. Association studies connecting consumption, addiction and side effects related to recreational cannabis use have led to the identification of several polymorphic genes that may play a role in the pharmacodynamics and pharmacokinetics of cannabis. Method: In total, 600 patients treated with cannabis were genotyped for several candidate polymorphic genes (single-nucleotide polymorphism; SNP), encoding receptors CNR1 and TRPV1; for the ABCB1 transporter; for biotransformation, bioactivation and biosynthesis; and CYP3A4, COMT and UGT2B7 conjugation. Results: Three polymorphic genes (ABCB1, TRPV1 and UGT2B7) were identified as being significantly associated with decline in pain after treatment with cannabis. Patients simultaneously carrying the most favourable allele combinations showed a greater reduction (polygenic effect) in pain compared to those with a less favourable combination. Considering genotype combinations, we could group patients into good responders, intermediate responders and poor or non-responders. Results suggest that genetic makeup is, at the moment, a significant predictive factor of the variability in response to cannabis. Conclusions: This study proves, for the first time, that certain polymorphic candidate genes may be associated with cannabis effects, both in terms of pain management and side effects, including therapy dropout. Significance: Our attention to pharmacogenetics began in 2008, with the publication of a first study on the association between genetic polymorphisms and morphine action in pain relief. The study we are presenting is the first observational study conducted on a large number of patients involving several polymorphic candidate genes. The data obtained suggest that genetic makeup can be a predictive factor in the response to cannabis therapy and that more extensive and planned studies are needed for the opening of new scenarios for the personalization of cannabis therapy.

## 1. Introduction

The therapeutic use of cannabis dates back to 2800 B.C. in China, and its diffusion in Europe to treat various diseases occurred mainly in the 18th and 19th centuries [1]. However, it was only in recent times that several complete and systematic studies on the therapeutic efficacy of cannabis were conducted, as well as extensive meta-analyses [2]. The increase in medical cannabis consumption revealed, alongside a number of beneficial effects, a variety of adverse side effects [3,4] and great inter-individual variability with a discouraging proportion (60–70%) of non-responders. This evidence stimulated research on the pharmacokinetics and pharmacodynamics of cannabis [5,6], which appears to be a difficult task, considering the vast number of cannabinoids existing in the phyto-complex of cannabis, the number exceeding 100, belonging to 18 different chemical classes, including terpenes, flavonoids and alkaloids [7], with quantities varying according to different brands and the extraction process. Furthermore, once they are absorbed, cannabinoids undergo massive hydroxylation by cytochrome P450, then pass on to glucuronate and are excreted [8] before active cannabinoid forms bind to various receptors, including CB1 and CB2, to exercise their main pharmacological effects. These intermediate forms can interact with other molecules in the phyto-complex and with the substrate, with potential synergistic and entourage effects [9]. The variability of phyto-complexes, together with patients’ biological diversity, may justify the vast inter-individual variability observed in clinical responses. Through association studies, the pharmacogenetic approach may represent an empirical approach to this problem, at least for what pertains to patients’ genetic variability. Association studies connecting consumption, addiction and side effects related to recreational cannabis use have led to the identification of several polymorphic genes that may play a role in the pharmacodynamics and pharmacokinetics of cannabis [10]. The vast functional differences of these genetic polymorphisms may justify part of the inter-individual variability observed in the medical use of cannabis as well. A pilot program for cannabis use guided by pharmacogenetics was already assessed, without, however, yielding any particular indications, most likely because of the low number of patients [11]. Stimulated by our previous experience on the pharmacogenetics of morphine in pain management [12], we envisaged a pilot study aimed at evaluating the feasibility of association analyses between the clinical effects of cannabis and certain genetic polymorphisms of patients undergoing chronic pain treatment. This may represent a new tool to better understand the action mechanisms of cannabis and its clinical effects. In total, 600 patients treated with cannabis were, therefore, genotyped for several polymorphic candidate genes (single-nucleotide polymorphism, SNP), encoding receptors CNR1 and TRPV1; for the ABCB1 transporter; for biotransformation, bioactivation and biosynthesis; and CYP3A4, COMT and Ugt2B7 conjugation. To the best of our knowledge, this is the first observational study conducted on a large number of patients involving different polymorphic candidate genes.

## 2. Methods

### 2.1. Study Design

An “open-label”, multi-centric, non-randomized observational study to assess the feasibility of possible associations between the outcomes of treatment with cannabis in patients suffering from chronic pain, for different diseases and certain polymorphic candidate genes.

### 2.2. Study Population

In total, 600 Italian patients of Caucasian origin, undergoing cannabis therapy for chronic pain, were recruited by Azienda USL Toscana Sud-Est, Zona Arezzo, San Donato Hospital (Tuscany, Italy), Department of Pain Medicine and Palliative Care, starting in 2018. The study was approved by the Tuscan Regional Ethical Committee (n. 1287) in 15 May 2018. None of the patients refused to take part in the project, reaching a total participation rate of 100%.

Patients were informed of the project and gave written consent to participating in the study and permitting their genotyping. They were interviewed and assessed for pain severity, in addition to collecting information on their age, sex and pain treatment. Exclusion criteria: age below 18 years, organic or psychiatric disorders interfering with judgement and comprehension abilities, persistence of pain for less than three months, no treatment with painkillers from traditional pharmacopoeia, pregnancy or breast-feeding, arrhythmia or ischemic cardiopathy, patients with severe psychiatric disorders, patients with a history of abuse or addiction to cannabis or other psychoactive substances.

Inclusion criteria: chronic pain, use of painkillers from traditional pharmacopoeia for at least three months without complete efficacy, full ability to understand.

Procedures were compliant with the ethical rules of the committee responsible for human experimentation and with the 1975 Helsinki Declaration, as reviewed in 2008.

Patients were grouped into five categories based on the type of disease originating the pain: (1) central nervous system diseases; (2) rheumatoid, arthritic, inflammatory and autoimmune diseases; (3) headache and migraine; (4) neuropathic pain; (5) cancer pain.

### 2.3. Patient Assessment

Interviews and data collection were carried out during the first examination (T0), after 4 weeks (T1) and after 12 months. During the study, a few patients decided to drop out of therapy, either because of unresponsiveness to cannabis therapy or because of relevant side effects. Measured variables were, mainly:(1)Pain intensity measured with the visual analogue scale (0–10);(2)Presence/absence of benefits (e.g., better sleep quality, muscle relaxation);(3)Presence/absence of side effects (e.g., brain fog, tachycardia, drowsiness);(4)Anxiety and depression symptoms, measured with the hospital anxiety and depression scale (HADS) [12].

### 2.4. Treatment with Cannabis

Administered cannabis preparations were: Cannabis FlosBedrocan^®^ (THC > 19%, CBD < 1%—Ministry of Health—The Netherlands), Cannabis FlosBedrolite^®^ (THC < 1%, CBD < 9%—Ministry of Health—The Netherlands), Cannabis FlosBediol^®^ (THC < 6,5%, CBD < 8%—Ministry of Health—The Netherlands), Cannabis FM2^®^ (THC 5–8%, CBD 7,5–12%—Military Pharmaceutical Chemical Institute Florence—Italy) and Cannabis FM1^®^ (THC 13–20%, CBD < 1%—Military Pharmaceutical Chemical Institute Florence—Italy). The different preparations of cannabis inflorescence were extracted in accordance with the SIFAP (Italian Association of Compound Pharmacists) method [13] and in accordance with the Legislative Decree 09 November 2015 of the Italian Ministry of Health, which prescribes preparing cannabis extracts according to the Good Compounding Practices FU. Because the decarboxylation reaction temperature for cannabinoids is approximately 110 °C [14], Cannabis FlosBedrocan, Cannabis FM2, Cannabis FM1 and Cannabis FlosBediol samples were heated at 115 °C for 70 min in a high-performance forced natural convection stove. Subsequently, cannabis samples were heated in a water bath at 100 °C for 40 min together with appropriate solvents (70 mg/mL in olive oil). The mixture of solvents and cannabis was then filtered. In order to prevent oxidation, 10 mg/mL of natural vitamin E FU and 0.2 mg/mL of butylhydroxytoluene (BHT) were added to the olive oil extract. The resulting organic phase was titrated for THC and CBD content and, subsequently, diluted in olive oil to obtain defined concentrations. The types of preparations and the quantities of THC and CBD administered daily were chosen according to the clinical needs assessed by the physician and used as variables in the subsequent statistical analyses.

### 2.5. Selection of Candidate Genes and Genotyping

Genetic variants (single-nucleotide polymorphism, SNP) in candidate genes were selected from previous publications with at least one positive association with a proven biological activity in individuals using cannabis for recreational purposes, given the lack of suitable indications in patients using cannabis for medical purposes. The 6 selected SNPs were: *MDR1/ABCB1* rs1045642; *TRPV1* rs8065080; *5-UGTB7* rs7438135; *CYP3A4* rs2242480; *CNR17B* rs1049353; *COMT* rs4680.

The DNA was obtained with an oral swab and extracted with DNA Extract All Lysis Reagent (Applied Biosystems Thermo Fisher Scientific, Milano, Italy), and SNPs were determined with TaqMan Assay (Applied Biosystem) with an RT-PCR One-Step-Plus system (Applied Biosystem by Thermo Fisher).

### 2.6. Statistical Analysis

The main dependent variable was pain reduction (∆VAS), calculated as the difference between basal pain and the pain assessed in each subsequent time. Each patient, therefore, also represented their own control. A multi-variate linear regression model (MLR) was used to assess the significance of the effect of variables on ∆VAS: year and centre of recruitment, age, gender, genotype, disease, painkillers and medication taken, type of cannabis and THC and CBD doses. The χ2 test and/or multi-nomial logistic regression were used for associations between categorical variables and the Kruskal–Wallis t (K–W) test for non-normally distributed variables. Associations with genotypes were initially tested within a co-dominant inheritance model (three separate genotypes). Simplified models were then applied: a dominant model (heterozygotes grouped with homozygotes for the minor allele) and a recessive model (heterozygotes grouped with homozygotes for the major allele) when effects were not significantly different. Being an explorative study, Bonferroni correction was not applied. All statistical analysis have been performed by Stat Graphics Centurion XVI, by Statgraphics Technologies, Inc. The Plains, VA, USA.

## 3. Results

Out of 600 patients recruited in the study, 564 were eligible for the analysis, as there was a complete database including correct genotyping. For the genotype quality control, a Hardy–Weinberg (H–W) equilibrium test was performed; all genotypes tested statistically in the H–W equilibrium (*p* < 0.05). The allelic proportions of the genes considered did not vary significantly in different disease groups or according to the different types of cannabis used. Patient demographics are reported in Table 1, with information on the medication and painkillers taken and pain perceived at the time of recruitment (VAS 0).

After the first two weeks, medical cannabis therapy was updated based on results, continuing over the following weeks and months. THC and CBD quantities significantly increased during the course of the study.

Pain reduction was approximately 20% in the first month, subsequently reducing by half in the third month and then further decreasing to 43% after one year. During the whole study, 443 patients (75.7%) dropped out of cannabis therapy, 179 of them putting forward various reasons, such as poor or no pain reduction, side effects or both, whereas the reasons for the remaining 264 were unknown. Polymorphism frequencies did not change significantly as a result of the drop-out factor. A decrease in ∆VAS during the first month was not significantly correlated with doses (squared) of THC (r^2^ = 0.091; b = 0.069; t = 0.73; *p* = 0.46) and of CBD (r^2^ = 0.29; b = 0.104; t = 1.08; *p* = 0.278), despite doses having doubled overall.

### 3.1. Association Study between Pain and Genotype

Physicians began modifying cannabis dosages starting from the second week of therapy in order to empirically personalize THC and CBD doses according to response and the cost/benefit ratio. The role of the polymorphic genes involved should have, therefore, progressively decreased due to both the cannabis dosage modifications and adaptation phenomena. The greatest percentage dosage increase, in fact, took place during the first month, as did the greatest percentage pain reduction. Furthermore, the systematic analysis of 18 studies on orally administered cannabis showed that the greatest pain reduction was detected in the first 2–8 weeks of treatment [2]. We, therefore, focused our analysis on the first month of treatment, also because the number of patients monitored decreased by only 6%. ∆VAS distribution is represented by the B&W graph in Figure 1. The great inter-individual variability within a continuous distribution was clear: certain patients (25%) reported an exacerbation of pain (negative or null ∆VAS), a further 25% reported very modest or null relief (∆VAS 0–1), 25% reported significant results (∆VAS > 1–3) and the remaining 25% reported acceptable or satisfactory pain relief (∆ > 3–10).

The multi-variate linear regression analysis for ∆VAS showed that, by applying the stepwise method, i.e., excluding variables with non-significant associations (*p* > 0.05), only polymorphisms for the *ABCB1* (Figure 2), *TRPV1* (Figure 3) and 5-*UBT2B7* (Figure 4) genes remained significantly associated with *p*-values of 0.0021, 0.0203 and 0.0178, respectively, considering the recessive model for each gene polymorphism.

Because the three genes did not show interactions and each patient was simultaneously characterized by the three polymorphisms of the three genes, to explore a possible polygenic effect, patients were grouped according to the eight possible combinations of polymorphisms, as reported in Table 2. Furthermore, our results showed that the three polymorphisms associated with the greatest decrease in pain were, respectively, CC + CT for *ABCB1*, CC for *TRPV1* and AA for *5-UGT2B7*. In Table 2, the eight combinations were, therefore, ordered according to the number of polymorphisms, which favoured a greater decrease in pain.

It was interesting to observe that as the number of favourable alleles in each genotype combination decreased, the efficacy of treatment with cannabis progressively decreased as well, suggesting an additive effect in each pair of alleles.

Patients were further grouped based on the amount of favourable allele combinations for each genotype: group one with two or three favourable alleles, group two with only one favourable allele and group three with no favourable alleles.

The MLR (multi-variate linear regression) analysis of ∆VAS showed a highly significant effect in new groupings (F = 9.89, *p* = 0.0001), as displayed by Table 3 and Figure 5, whereas no other factor considered was even close to significance yet, including the year of recruitment, gender, disease, medication and other painkillers taken, nor the age and quantity of cannabis administered. Differences between the three groups were significant (*p* < 0.05).

Patients were further grouped based on the amount of favourable allele combinations for each genotype: group one with two or three favourable alleles, i.e., CC + CT for *ABCB1*, CC for *TRPV1* and AA for *UGT2B7*; group two with only one favourable allele and group three with no favourable alleles, i.e., TT for *ABCB1*, TT + CT for *TRPV1* and GG + AG for *AGT2B7*. More conservative statistical analyses showed that the two most respondent groups were not different from each other, while the difference between them and the least responsive group remained significant (*p* < 0.05). Consequently, we concluded that the poor responders to cannabis were patients carrying three non-favourable alleles.

Patients grouped on the basis of their genotypes being more or less favourable to a pain decrease could be classified in groups with statistically significantly (*p* < 0.05) different pain decreases. Group three, therefore, showed a 0.56 ∆VAS, classifying Group three as non-responders. Group two showed a 1.29 ∆VAS—that is, 2.3 times higher than group three—and could be considered to constitute responders. Group one showed a 1.73 ∆VAS—that is, 3.1 times higher than group three—and could be considered to have the highest number of responders.

At the beginning of the study, 98 patients presented evidence of depression, and this condition was significantly associated with *CNR1* polymorphism, rs1049353 (chi squared 6.12, *p* = 0.013), with a larger proportion of women (chi 5.4; *p* = 0.02; odds ratio 1.8 CL 1.1–3.1).

### 3.2. Drop Out

In total, 78 patients dropped out of treatment with cannabis during the first month for various reasons, mainly for the lack of pain reduction, side effects or both. The average pain reduction in drop-out patients was null or negative (∆VAS = −0.17) compared to the others (∆VAS = 1.7), and the difference was highly significant (K–W, t = 77.8, *p* = 0).

The χ2 test showed a strongly significant association (χ2 = 31.31, *p* < 0.0000) between the drop-out patient group and those suffering from multiple adverse events, such as tachycardia, drowsiness, dry mouth and hyperactivity. Furthermore, there was a significant association (χ2 = 5.6 *p* < 0.018) between the drop-out phenomenon and the polymorphism (recessive model) of the gene *CNR1* (*CB1*) rs1049353. The odds ratio for patients with the GG + GA genotypes vs. AA homozygotes was 2.4 (CL 1.1–5.2), indicating that the former were 2.4 times more likely to continue treatment with cannabis than AA homozygotes. Logistic regression confirmed the role of *CNR1* polymorphism in the risk of treatment drop-out for AA homozygotes, excluding significant associations with all the other possible factors we tested.

### 3.3. Side Effects

Regarding depression, in T0, 98 patients were affected by symptoms attributable to a depressive disorder. At the end of the first month, 25 patients had recovered, while 12 new patients with symptoms attributable to a depressive disorder were added to the group. Given the low number, statistical analyses were not performed.

In respect to anxiety, the logistic analysis showed significant associations between the presence of anxiety and disease (*p* = 0.01), as group two was more strongly associated with anxiety and the use of medication (*p* = 0.0002). Those who took medication were at higher risk, probably because they were affected by more serious diseases.

## 4. Discussion

In this study, we showed that, despite the complexity of cannabis pharmacology and the difficulty in precisely quantifying the results of treatment with cannabis, it is possible to assess the role of certain genetic polymorphisms in patients and to explain part of the great inter-individual variability observed in pain reduction. On average, the pain reduction in remaining patients (non-drop-out) at the end of the first month amounted to approximately 20%, but with great variability. Part of this was significantly associated with three polymorphic candidate genes: *ABCB1*, *TRPV1* and *UGT2B7*. The three other genes, as the other variables (year and centres of recruitment, age, sex, disease, concomitant use of other painkillers and medication to treat the patient’s main disease, type of cannabis and quantity of THC and CBD taken) not only showed no significant association, but remained very far from significance. It is very likely that the associations observed were in part disguised by the constant adjustment of cannabis dosages, being approximately doubled on average, but reduced in certain cases, and by the fact that approximately 30% of patients changed their type of cannabis during the first month of treatment to the “titrate” type.

The rs1045642 polymorphism, also known as *C3435T*, of the *MDR1/ABCB1* gene, which encodes a transporter protein (MRP1), produces a “silent” missense variant, which, however, determines the reduction in the expression of the transporter protein [15]. Patients with neuropathic pain treated with nortriptyline and TT homozygotes showed a lower pain reduction than others [16]. In this study, homozygous TT patients were less responsive to treatment with cannabis than the others. These results were in accordance with those obtained in a morphine treatment for chronic pain, as TT homozygous patients were less responsive than the others [12], confirming the role of this gene in the control of drug transport. Furthermore, these results were supported by the discovery that, in heavy cannabis users, THC plasma levels were significantly higher in carriers of at least one T allele [17]. An increase in the risk of neuropathy has also been reported after treatment with taxanes in TT homozygous patients [18]. It is interesting to note that this polymorphism has also been associated with variability in the cold pain threshold and in pain tolerance in healthy males [19], suggesting a role of the *MDR1/ABCB1* gene in pain control, the pharmacological treatment aside.

The *TRPV1* polymorphism (rs8065080), also known as c.1191A > G, is a common variant in the *TRPV1* gene (transient receptor potential cation channel), which encodes, among others, vanilloid receptor subtype-1 and the capsaicin receptor. The polymorphism determined by rs8065080 was associated with salt sensitivity [20], the tolerance of cold, heat, pinpricks, homozygous carriers of the C allele with a higher pain tolerance [21] and with painful osteoarthritis of the knee [22]. In our study, CC homozygotes experienced a significantly greater pain reduction compared to the other genotypes. There is extensive evidence of the role of the TRPV1 gene in the therapeutic activity of cannabis. It has recently been demonstrated that TRPV1 channels are activated by cannabinoids [23], as by myrcene, contained in cannabis, which is recommended to obtain analgesic effects [24]. Cannabis has been successfully used in the treatment of epilepsy, and the inhibition of the transient receptor potential of vanilloid type 1 channels could be one of the mechanisms at the basis of CBD anti-seizure effects [25,26].

The *UGT2B7* gene encodes UDP-glucuronosyltransferase-2B7, involved in phase II metabolism, conjugating a variety of compounds such as analgesics (morphine), carboxylic non-steroidal anti-inflammatory drugs (ketoprofen) and anti-cancer drugs (all-trans retinoic acid) [27]. The rs7438135 polymorphism has been proven to be involved in opioid withdrawal symptoms [28] and in the glucuronidation model of morphine [29]. In the present study, AA homozygous patients reported a significantly greater pain reduction compared to AG and GG heterozygotes. These results were supported by the evidence that oncological patients with an AA genotype are, generally, more sensitive (less tolerant) to opioids compared to those with AG genotypes [30].

The fact of having found three polymorphic genes which individually significantly influence pain decrease after treatment with cannabis poses the problem of the impact of their possible combinations in patients. As expected, patients simultaneously carrying the most favourable allelic combinations showed a greater pain reduction (∆VAS = 1.73) compared with those with a less favourable combination (∆VAS = 0.56). Considering genotype combinations, we could, therefore, group patients into good responders, intermediate responders and poor or non-responders. We are aware that this study included many variables, such as cannabis types, contents of active principles, different groups of diseases and patients’ overall clinical situations, the interactions of which were difficult to assess as they were not planned a priori, but could have important effects on the assessment of pain reduction. However, the results suggested that a patient’s genetic makeup is, at the moment, a significant descriptive factor of variability in the response to cannabis. Changes in cannabis dosages, doubled on average, which occurred during the first month and were empirically guided, were not associated with any factor, including genetic factors, thus, suggesting that the initial dosage, apparently low, was due to the caution of physicians who administered a drug which was not well known to patients who were treated for the first time. Furthermore, the association study showed that the A allele of the *CNR1* gene, rs1049353, was a therapy drop-out risk factor. This gene encodes the most relevant cannabinoid receptor, the polymorphism of which was extensively studied in depression therapy [31,32].

## 5. Conclusions

This study proved, for the first time, that certain polymorphic candidate genes may be associated with cannabis effects, both in terms of pain control and side effects, including therapy drop-out. This approach is, therefore, feasible and, if adequately developed, could possibly contribute to shedding light on the complex pharmacokinetics and pharmacodynamics of cannabis, making physicians more confident in its therapeutic use. We are aware that only by extensively expanding the polymorphic genes panel and the number of their polymorphisms could it be possible to hypothesize a more personalized therapy, aided by the knowledge of patients’ genetic characteristics. It is our opinion that an association study on the whole genome (GWAS, genome-wide association study) could lead to important progress in this field.

## Figures and Tables

**Figure 1 genes-13-01832-f001:**
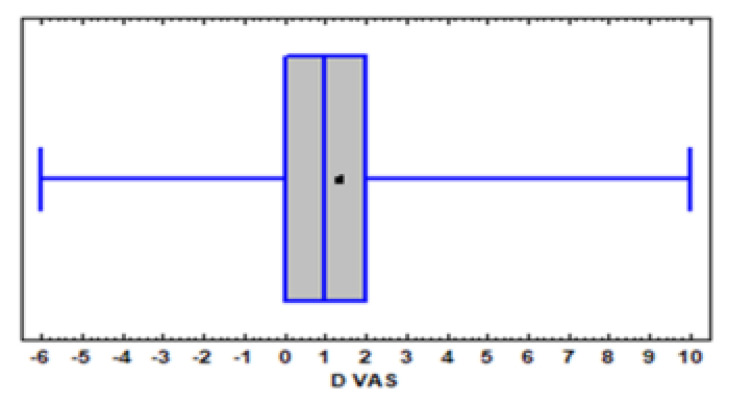
**DVAS** Box and whisker plot of ∆VAS score distribution after a month of treatment with cannabis. The average ∆VAS value is represented by the X.

**Figure 2 genes-13-01832-f002:**
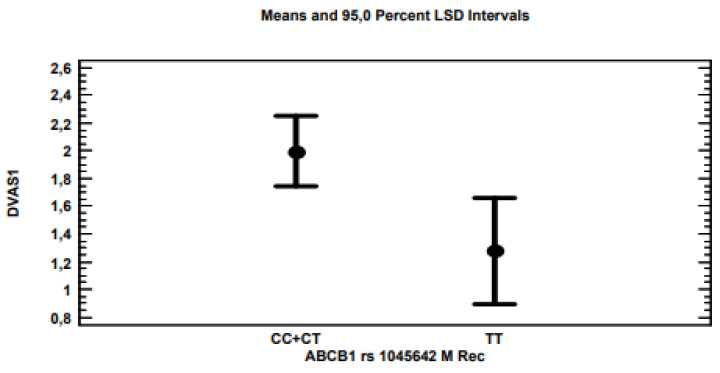
***ABCB1*** The analysis showed that CC homozygous patients and CT heterozygotes for the *ABCB1* gene were associated with an average pain reduction of approximately 2 VAS points, compared to the 1.3 VAS points of TT homozygotes.

**Figure 3 genes-13-01832-f003:**
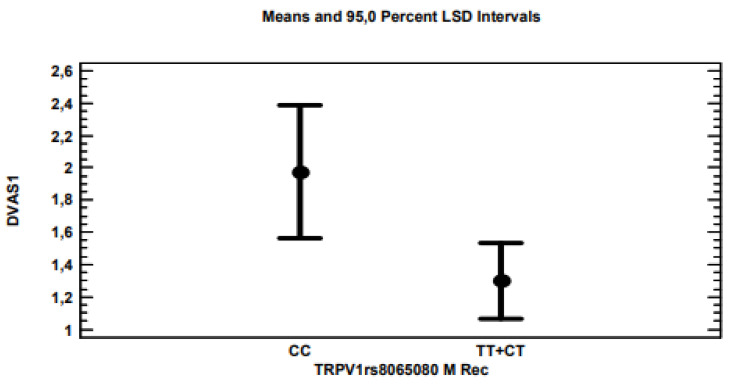
***TRPV1*** CC homozygous patients for the *TRPV1* gene obtained an average pain decrease of 2 VAS points, compared to the 1.3 VAS points of TT homozygotes or CT heterozygotes.

**Figure 4 genes-13-01832-f004:**
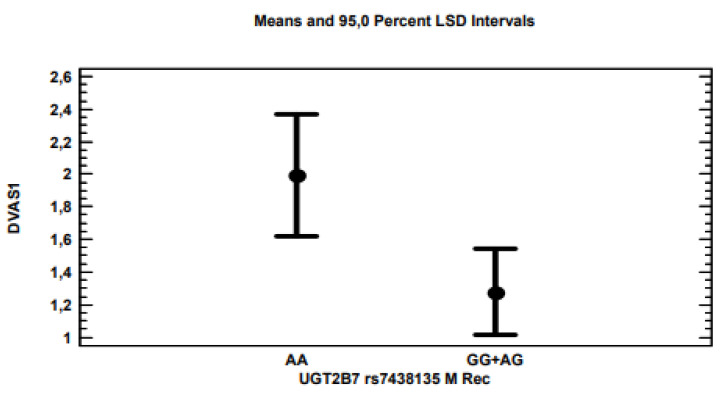
***UBT2B7*** AA homozygous patients for the *UGT2B7* gene reported an average pain decrease of 2 VAS points, compared to the 1.3 VAS points of GG homozygotes or AG heterozygotes.

**Figure 5 genes-13-01832-f005:**
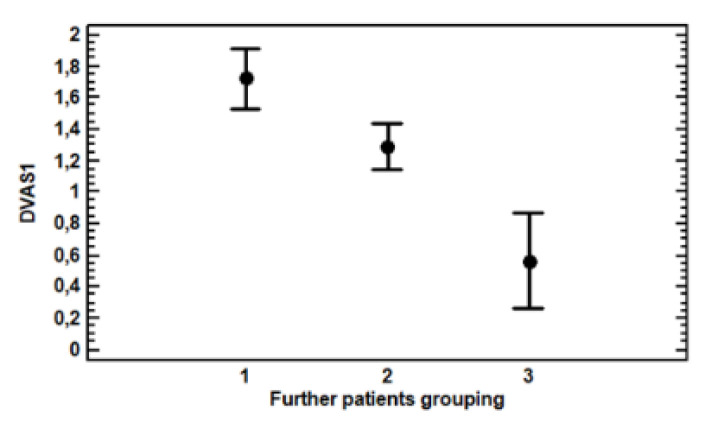
Plot of average values of ∆VAS and standard error of the three groups of patients grouped according to the number of alleles conducive to pain reduction after one month of treatment with cannabis.

**Table 1 genes-13-01832-t001:** Sex, average age, percentage of patients taking medication and analgesics and VAS score at time 0.

Gender	n. Patients	Age ± s.d	Drugs%	Analgesics%	VAS 0 ± s.d
Female	400	56.8 ± 16.4	25.2	55.6	7.6 ± 1.8
Male	164	56.9 ± 17.9	39.2	42.0	7.4 ± 1.7
Total	564	56.8 ± 16.8	29.4	51.5	7.6 ± 1.7

**Table 2 genes-13-01832-t002:** Genotypes groupings based on allelic combinations and ordered by the number of favourable combinations for pain reduction.

Combination of	n. Patients	n. Favourable			Genotypes	∆VAS ± s.d
Genotypes		Alleles	ABCB1	TRPV1	UGT2B7	
1	18	3	CC + CT	CC	AA	2.1 ± 2.3
2	5	2	TT	CC	AA	2.2 ± 3.6
3	98	2	CC + CT	TT + CT	AA	1.6 ± 1.8
4	63	2	CC + CT	CC	GG + AG	1.6 ± 2.4
5	274	1	CC + CT	TT + CT	GG + GA	1.3 ± 1.8
6	32	1	TT	TT + CT	AA	1.2 ± 1.9
7	19	1	TT	CC	GG + AG	1.2 ± 1.5
8	76	0	TT	TT + CT	GG + AG	0.6 ± 1.5

**Table 3 genes-13-01832-t003:** Average ∆VAS of patient groups constituted on the basis of the number of favourable alleles for pain decrease after treatment with cannabis.

Group	n. Patients	∆VAS ± s.d
1	177	1.72 ± 2.11
2	316	1.29 ± 1.79
3	71	0.56 ± 1.55

## Data Availability

The data presented in this study are available on request from the corresponding author. The data are not publicly available because we are evaluating an eligible publicly accessible repository.

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
