# Peer review of "The Pharmacogenetics of Cannabis in the Treatment of Chronic Pain"

_genes, 2022, doi:10.3390/genes13101832_

Round 1
Reviewer 1 Report
This manuscript describes the association between the presence of candidate genetic variants within genes involved in the metabolism of drugs and encoding receptors CNR1 and TRPV1 in an observational study conducted in 600 patients treated with cannabis for the treatment of chronic pain. The evidence supporting this assertion is based on a clear strategy for the genotyping of several candidate polymorphic genes finding three polymorphic genes (ABCB1; TRPV1 and UGT2B7) as significantly associated with a decline in pain after treatment with cannabis. Moreover, the authors grouped cannabis treated patient's cohort in three groups of patients according to the most favourable allele combinations identifying a genetic make-up related to a group of patients in which there was a greater reduction (polygenic effect) in pain compared to those with a less favourable combination. The manuscript has numerous information and is of good quality to be published but I would suggest applying bonferroni's correction in the analysis of data to put in evidence the power estimation of results in this association study.
Moreover, the classification of patient groups in high responders, responders and non-responders could correlate with any polymorphic variants conditioning the phenotype of ultra-, extensive- or intermediate/poor- metabolizer: do authors have explored this aspect?
On the line 232-236 there is a reference to table 3 and fig.5. In table 3 are reported the 3 patient groups based on the combination of almost favorable alleles while fig. 5 is lacking. Please add fig. 5 to the manuscript and explain better the results obtained by MLR analysis.
Please correct the mistake at line 60 replacing UBTB7 with Ugt2B7
Author Response
Good morning, attached you will find thye manuscript with the fig.5 added and better explained and the clarifications required
Best Regards

Reviewer 2 Report
The present study investigates the possible relationship between several polymorphic genes and the efficacy of cannabis in the treatment of chronic pain. The explored topic is of considerable scientific and clinical interest. The paper could represent an interesting contribution to the research field. I have no substantive comments on the views expressed by the authors. I only advise the authors to make some formal corrections:
1. Check punctuation in the abstract section and in the main sections of the article (eg. Line 287);
2. Keywords should be expanded to facilitate access to this article;
3. References number 2, 25 and 26 are not noted as indicated by the journal;
Thank you for considering my opinion.
Author Response
Good morning,
I send you the manuscript with the correction you suggested
Best Regards
